# Quantitative Proteomics of Maternal Blood Plasma in Isolated Intrauterine Growth Restriction

**DOI:** 10.3390/ijms242316832

**Published:** 2023-11-27

**Authors:** Natalia L. Starodubtseva, Alisa O. Tokareva, Maria V. Volochaeva, Alexey S. Kononikhin, Alexander G. Brzhozovskiy, Anna E. Bugrova, Angelika V. Timofeeva, Evgenii N. Kukaev, Victor L. Tyutyunnik, Natalia E. Kan, Vladimir E. Frankevich, Evgeny N. Nikolaev, Gennady T. Sukhikh

**Affiliations:** 1National Medical Research Center for Obstetrics, Gynecology and Perinatology Named after Academician V.I. Kulakov of the Ministry of Healthcare of Russian Federation, 117997 Moscow, Russia; alisa.tokareva@phystech.edu (A.O.T.); m_volochaeva@oparina4.ru (M.V.V.); konoleha@yandex.ru (A.S.K.); agb.imbp@gmail.com (A.G.B.); anna.bugrova@gmail.com (A.E.B.); avtimofeeva28@gmail.com (A.V.T.); e_kukaev@oparina4.ru (E.N.K.); vtyutyunnik@emcmos.ru (V.L.T.); n_kan@oparina4.ru (N.E.K.); v_frankevich@oparina4.ru (V.E.F.); g_sukhikh@oparina4.ru (G.T.S.); 2Moscow Institute of Physics and Technology, 141700 Moscow, Russia; 3Emanuel Institute of Biochemical Physics, Russian Academy of Sciences, 119334 Moscow, Russia; 4V.L. Talrose Institute for Energy Problems of Chemical Physics, N.N. Semenov Federal Research Center for Chemical Physics, Russian Academy of Sciences, 119334 Moscow, Russia; 5Laboratory of Translational Medicine, Siberian State Medical University, 634050 Tomsk, Russia

**Keywords:** intrauterine growth restriction, pregnancy, plasma, quantitative proteomics, diagnostics, prognosis, mass spectrometry

## Abstract

Intrauterine growth restriction (IUGR) remains a significant concern in modern obstetrics, linked to high neonatal health problems and even death, as well as childhood disability, affecting adult quality of life. The role of maternal and fetus adaptation during adverse pregnancy is still not completely understood. This study aimed to investigate the disturbance in biological processes associated with isolated IUGR via blood plasma proteomics. The levels of 125 maternal plasma proteins were quantified by liquid chromatography-multiple reaction monitoring mass spectrometry (LC-MRM MS) with corresponding stable isotope-labeled peptide standards (SIS). Thirteen potential markers of IUGR (Gelsolin, Alpha-2-macroglobulin, Apolipoprotein A-IV, Apolipoprotein B-100, Apolipoprotein(a), Adiponectin, Complement C5, Apolipoprotein D, Alpha-1B-glycoprotein, Serum albumin, Fibronectin, Glutathione peroxidase 3, Lipopolysaccharide-binding protein) were found to be inter-connected in a protein–protein network. These proteins are involved in plasma lipoprotein assembly, remodeling, and clearance; lipid metabolism, especially cholesterol and phospholipids; hemostasis, including platelet degranulation; and immune system regulation. Additionally, 18 proteins were specific to a particular type of IUGR (early or late). Distinct patterns in the coagulation and fibrinolysis systems were observed between isolated early- and late-onset IUGR. Our findings highlight the complex interplay of immune and coagulation factors in IUGR and the differences between early- and late-onset IUGR and other placenta-related conditions like PE. Understanding these mechanisms is crucial for developing targeted interventions and improving outcomes for pregnancies affected by IUGR.

## 1. Introduction

Intrauterine growth restriction (IUGR) occurs when a fetus doesn’t reach its genetically determined growth potential [1]. Every year, nearly 21 million babies are born smaller than expected for their gestational age [2]. IUGR remains a significant concern in modern obstetrics, linked to high neonatal health problems and even death, as well as childhood disability, affecting adult quality of life [3,4]. Major obstetric issues, like preterm birth, pre-eclampsia (PE), and IUGR, pose serious challenges in modern obstetrics. They lead to negative outcomes and present global medical and socioeconomic challenges.

Pre-eclampsia (PE) is one of the leading causes of maternal and perinatal morbidity and mortality, accounting for one-fifth of maternal deaths worldwide. It has been estimated that every 12 min, one woman dies globally from PE and eclampsia. This complication during pregnancy increases the risk of fetal growth restriction and the frequency of iatrogenic preterm births. Women who have experienced PE during pregnancy face a 2–4 times higher risk of developing heart failure and dying from cardiovascular diseases later in life compared to those with pregnancies without hypertensive complications [5]. Pre-eclampsia and fetal growth restriction develop due to impaired trophoblast invasion and inadequate remodeling of spiral arteries. Based on this, it has been suggested that these pregnancy complications should be grouped into one category—ischemic placental disease [6].

Although conditions like PE and IUGR often coexist, recent data suggest that they have distinct causes. According to the Huppertz hypothesis, a failure in the differentiation of the extravillous trophoblast pathway leads to IUGR, unlike PE, which may result from a failure in the differentiation of the villous trophoblast pathway with its typical characteristics, such as systemic changes in maternal endothelium and inflammatory response [7]. Therefore, it is vital to study not only their combination, presenting 12–35% of cases, but also isolated PE and IUGR [8]. However, most previous studies combine IUGR with severe PE, making it challenging to determine molecular features of fetal growth restriction [9,10].

One significant challenge in studying IUGR is the lack of a definitive diagnostic standard [11]. Baby birth weight is a key factor in diagnosing IUGR. However, setting a specific weight threshold can misclassify affected and unaffected babies. Doppler flow velocimetry of the utero-placental and fetal circulations is also used for classification. While ultrasound examinations were once considered the “gold standard” for diagnosis, they often overestimate fetal weight, with IUGR confirmed postnatally in 75% of cases. Currently, antenatal and postnatal IUGR diagnoses align in only 12–47% of pregnancies, with around 10% of results being false positives. This highlights the ongoing challenge of verifying IUGR during pregnancy. The primary diagnostic approach involves combining ultrasound exams with Doppler parameter assessments. In 2016, international consensus criteria within the Delphi protocol divided IUGR into early and late types based on clinical, ultrasound, and Doppler indices [12].

Early IUGR, manifesting before 32 weeks, is linked to issues in implantation and placental development, severe cardiovascular adaptations, and high fetal tolerance to oxygen deprivation. This results in higher perinatal health problems and death rates. Late IUGR involves milder placental problems, less centralized blood circulation, lower fetal tolerance to oxygen deprivation, and lower perinatal health problems and death rates compared to the early type [13]. Most authors agree that managing early IUGR poses the most significant challenge while diagnosing the late type is more important.

Proper placental development relies on various factors such as angiogenic growth factors, hormones, transcription factors, cytokines, and cell adhesion molecules [14]. Disruptions in the remodeling of spiral arteries lead to placental dysfunction [15]. There is considerable overlap between the pathogenesis of pre-eclampsia and IUGR. Nevertheless, the connection between abnormal placentation, maternal predisposition, disease onset and severity, and the health outcomes of newborns is not well understood [16,17].

The question of whether small for gestational age (SGA) infants constitute an independent medical category is debated. SGA is defined as babies with a birth weight below the 10th percentile for their gestational age [18]. Infants with SGA fall into two main groups: constitutionally normal infants who are genuinely SGA and infants with IUGR. SGA and IUGR are often used interchangeably, but this is incorrect. SGA cannot reliably indicate IUGR because some IUGR infants may have a birth weight above the 10th percentile [19,20]. Nevertheless, there is a clear link between birth weight, fetal growth rates, and adverse perinatal outcomes [21]. Distinguishing between SGA and IUGR remains a challenging task [14].

The most straightforward way to discover new biomarkers for IUGR is by analyzing maternal blood samples. However, current research in transcriptomics (RNA, microRNA), epigenetics (gene methylation), lipidomics, metabolomics, and proteomics for IUGR diagnostics has made limited progress [20]. In recent years, some molecules in maternal blood have been identified as potentially linked to IUGR’s development. These include beta-human chorionic gonadotrophin (β-hCG), pregnancy-associated plasma protein-A (PAPP-A), vascular endothelial growth factor (VEGF), placental growth factor (PlGF), and soluble endoglin (sEng), A Disintegrin and Metalloprotease 12 (ADAM12), placental protein 13 (PP13), alpha-fetoprotein (AFP), inhibin A, activin A, follistatin, placental growth hormone (PGH), neural cell adhesion molecule (N-CAM), fibroblast growth factor (FGF), Insulin-like growth factor-I (IGF-I), and Insulin-like growth factor binding proteins (IGFBP-1, IGFBP-3, and IGFBP-4). One commonly used approach in practice is assessing the sFlt-1/PlGF ratio during pregnancy screening (between weeks 11 to 14), which can predict the risk of developing IUGR with a sensitivity of 63% (95% CI: 54–71) and a specificity of 84% (95% CI: 83–85). However, the positive likelihood ratio remains low at 3.55 (95% CI: 1.98–6.34) [9].

Utilizing quantitative proteomics to discover non-invasive diagnostic markers for IUGR holds promise in obstetrics. Since the development of a systemic inflammatory response, coupled with vascular and hemostatic abnormalities, is a key factor in IUGR’s pathogenesis, quantifying specific proteins associated with these aspects as biological markers becomes highly relevant. Multiple reaction monitoring (MRM) is one of the most advanced tools of mass spectrometry (MS) based proteomics used for biomarker discovery and validation [22]. Rigorously validated according to CPTAC guidelines, kits allow for the quantification of hundreds of plasma/serum proteins in a small sample volume (about 10 µL). The proteins of the BAK 125 kit (MRM Proteomics Inc., Montreal, QC, Canada) cover a wide range of potential disease biomarkers, including diabetes, cancer, neurodegenerative, autoimmune, cardiovascular, renal, eye, bone, chronic kidney diseases, and pregnancy complications [23]. Among them, 44 proteins are FDA-approved biomarkers [24], and 51 proteins are proposed as cardiovascular disease biomarkers [25].

Early and non-invasive diagnosis of IUGR and SGA is crucial to reduce neonatal health problems and deaths. Clarification of the molecular mechanisms associated with the disruption of adaptation of the mother and fetus during pregnancy will enable the identification of new early predictors of dangerous complications of pregnancy, as well as potential targets for prevention and therapy. In this study, quantitative analysis of 125 maternal plasma proteins was used for the first time using liquid chromatography-multiple reaction monitoring mass spectrometry (LC-MRM MS) to investigate the pathological molecular mechanisms underlying a rare and dangerous pathology of pregnancy, isolated IUGR.

## 2. Results

### 2.1. Features of Pregnancy Course and Outcomes with IUGR

The study involved 50 pregnant women from five groups: normotensive pregnancies complicated by early/late IUGR, the comparison group with an SGA fetus and control pregnancies (before and after 32 weeks) with an appropriately grown fetus. Table 1 presents the demographic and clinical data of pregnant women and fetuses participating in the study. Groups were matched on the mother’s age or BMI, nicotine addiction, and somatic and gynecological disorders. Additionally, the women in the study did not have a history of IUGR. Early IUGR manifested at 25 ± 3 weeks; late IUGR—at 34 ± 2 weeks; and SGA—at 35 ± 2 weeks.

Antenatal diagnosis of early fetal growth restriction (IUGR) was achieved in 90% of cases (9 out of 10). In the late IUGR group, it was established in 80% of cases, and in the SGA group—in 70% of cases. Notably, in 30% of SGA cases (3 out of 10), a late form of IUGR was antenatally diagnosed, highlighting a common disparity between antenatal and postnatal diagnoses of IUGR [10].

The newborn sex was comparable between groups (*p* = 0.891). The course of the early neonatal period in group 1 was significantly more often complicated by respiratory distress syndrome (RDS) and pulmonary hemorrhage compared to groups 2–5 (*p* = 0.016, *p* = 0.002, respectively). Intraventricular hemorrhage (IVH) was more common in groups 1, 2, and 3 (*p* = 0.024), and disseminated intravascular coagulation (DIC) occurred in half of the newborns of group 1 (*p* = 0.023). The detailed data is presented in Appendix A.

### 2.2. Quantitative Analysis of Plasma Proteins in Maternal Blood

A targeted quantitative analysis of 125 proteins in the blood plasma of pregnant women was performed by LC-MRM MS with corresponding internal stable isotope-labeled peptide standards (SIS). The studied 125 plasma proteins are the major and moderately represented blood proteins, accounting for more than 99% of the plasma protein mass. The dynamic range of these proteins spans 5.43 orders of magnitude (Appendix A). Most BAK 125 kit proteins were proposed as markers for cardiovascular, cancer, and neurodegenerative diseases [26,27,28,29].

As pregnancy advances, there are notable changes in the molecular and cellular composition of blood plasma. We observed a statistically significant (*p* < 0.05) decrease in the levels of nine proteins during later stages of pregnancy (late control vs. early control): Apolipoprotein A-IV, Cathelicidin antimicrobial peptide, Complement factor H, Fetuin-B, Haptoglobin, Plasma protease inhibitor, Plasminogen, Retinol-binding protein, and Transthyretin. Additionally, there was an increase in the level of Protein S100-A9 (Appendix A).

In total, thirteen proteins exhibited statistical differences between the control and IUGR groups at an uncorrected *p*-value of < 0.05 (Appendix A). Most of these proteins increased with IUGR. Gelsolin and Alpha-2-macroglobulin exceeded the 5% cutoff, and Apolipoprotein A-IV surpassed the 10% cutoff after the Benjamini–Hochberg multiple test correction. Nine proteins had a higher-than-medium effect size (Cohen’s d values). Apolipoprotein B-100 and Apolipoprotein(a) were crucial for distinguishing the control group, while the levels of Alpha-2-macroglobulin, Apolipoprotein A-IV, Adiponectin, Complement C5, Apolipoprotein D, Alpha-1B-glycoprotein, and Serum albumin appeared to be most important for distinguishing IUGR (Figure 1). Nevertheless, these differences were insufficient to establish a reliable method for distinguishing between the two conditions (control vs. IUGR). Consequently, all the 125 quantified proteins were considered as features for correlation analysis with clinical data and building a diagnostic binary classifier.

A more detailed analysis unveiled distinctive features in the maternal plasma proteome across IUGR types and SGA. In early IUGR, there was a statistically significant increase in maternal blood plasma levels of Adiponectin (ADIPOQ), Alpha-2-macroglobulin (A2M), Gelsolin (GSN), Pregnancy zone protein (PZP), Protein S100-A9 (S100A9), Serum albumin (ALB). Conversely, there was a decrease in Apolipoprotein B-100 (APOB100), Cathelicidin antimicrobial peptide (CAMP), Complement component C7 (C7), and Tissue-type plasminogen activator (PLAT) (Figure 2A). For late IUGR, there was an increase in plasma levels of Apolipoprotein A-IV (APOA4), Biotinidase (BTD), Fibrinogen gamma chain (FGC), Gelsolin (GSN), Glutathione peroxidase 3 (GPX3), Plasminogen (PLG), and Retinol-binding protein 4 (RBP4). Conversely, there was a decrease in Cation-independent mannose-6-phosphate receptor (CI-MPR) (Figure 2B). The most notable changes in the protein composition of maternal plasma in late IUGR compared with low birth weight fetuses were observed for Gelsolin (GSN), Serum paraoxonase/lactonase 3 (PON3), Keratin type I cytoskeletal 10 (KRT10), Apolipoprotein C-I (APOC1), Hemopexin (HPX), and Transthyretin (TTR) (Appendix A).

We identified statistically significant correlations (*p* < 0.05) between blood plasma proteins and several maternal, fetal and newborn clinical characteristics. These correlations included the timing of the IUGR onset, the severity of the pathology (Figure 3A), the timing of delivery, the length of stay of newborns in the intensive care unit, changes in fetoplacental blood flow, the newborn Apgar score at 1st and 5th minute, newborn weight (Figure 3B), the umbilical cord blood pH, lactate, glucose levels, and the development of IVH in children (Appendix A).

Several maternal plasma proteins could be proposed as prognostic markers of newborn hemorrhage in the early neonatal period, including pulmonary, gastric, and intraventricular bleeding, as well as disseminated intravascular coagulation. These proteins included Apolipoprotein A-IV, Pregnancy zone protein, and Serum paraoxonase/lactonase 3 (Appendix A).

GO pathway analysis revealed a broad range of processes affected by IUGR (Figure 4), encompassing plasma lipoprotein (chylomicrons, HDL, LDL) assembly, remodeling, and clearance; lipid metabolism (particularly, cholesterol and phospholipids); hemostasis (including platelet degranulation); and immune system regulation (Appendix A). Notably, early IUGR is associated with disturbances in complement and coagulation cascades, as well as plasma lipoprotein assembly, remodeling, and clearance. Conversely, late IUGR is strongly linked to responses related to lipid hydroperoxide, vitamin digestion and absorption.

### 2.3. Binary Classifiers Based on Plasma Proteins Level in Maternal Blood

Based on the results of a quantitative proteomic analysis of 125 maternal blood plasma proteins at delivery, three diagnostic models were developed using the logistic regression method.

Model “1” was created for the diagnosis of early IUGR, incorporating Alpha-2-macroglobulin as a variable. This model demonstrated a sensitivity of 90% and a specificity of 90% (Table 2, Appendix A).

Similarly, models “2” and “3” were developed for diagnosing late forms of IUGR and for the differential diagnosis of IUGR and SGA (Table 2, Appendix A).

It is essential to anticipate the development of hypoxia in the fetus and predict perinatal asphyxia to make informed decisions regarding surgical delivery. A logistic regression model was developed for predicting asphyxia using maternal Kallistatin and Metalloproteinase inhibitor 2, yielding an AUC of 0.93, sensitivity of 100%, and specificity of 86% (Appendix A).

To forecast the newborn’s hemorrhage, a logistic regression model was constructed based on the levels of maternal plasma Hemoglobin subunit alpha, Protein S100-A9, Fibrinogen beta chain, Serum paraoxonase/lactonase 3, Alpha-1-acid glycoprotein 1, and Lipopolysaccharide-binding protein. This model achieved a sensitivity of 92% and specificity of 76% (Figure 5).

## 3. Discussion

IUGR stands as one of the primary contributors to early neonatal morbidity and mortality across the globe and is responsible for 30% to 70% of all stillbirths [30]. It is closely linked to an increase in iatrogenic preterm births, which, in turn, drives up the incidence of premature births [31,32,33]. According to a study by Fatima Crispi et al. in 2018, individuals born with a weight below the 5.5th percentile face twice the risk of cardiovascular disease-related death before the age of 65, alongside a heightened susceptibility to metabolic and neurological complications [34]. IUGR represents a syndrome with multiple causative factors, characterized by the development of placental abnormalities and disruptions in placental processes that reduce nutrient delivery to the fetus and cause tissue hypoxia. As stated by Malhotra et al. in 2019, up to 50% of IUGR cases are undetected [31]. The most pressing issue in managing IUGR is not only achieving effective prediction, diagnosis, and monitoring but also providing antenatal assessments of fetal well-being to facilitate timely delivery and reduce the rate of unnecessary premature births.

In 30% of cases, early IUGR is accompanied by severe PE. Either isolated PE, isolated IUGR, or combined PE and IUGR can develop. The pathophysiological mechanisms underlying IUGR are intricate and diverse. A previous study of ours highlighted the distinct pathogenetic profiles of PE and IUGR [8]. Our current study concentrates on isolated IUGR without the influence of PE as a separate placenta-associated condition. An absolute quantitative targeted proteomics analysis (LC-MRM MS) included 125 maternal blood plasma proteins proposed as markers for more than 10 pathologies [23]. The research demonstrates distinct maternal plasma proteomic patterns for early and late IUGR and SGA, compared to control groups matched for gestational age. This represents the first quantitative investigation of a wide array of maternal circulating proteins and associated pathophysiological pathways, offering potential diagnostic/prognostic markers and therapeutic targets.

During the progression of pregnancy, nine plasma proteins exhibited statistically significant decreases, while Protein S100-A9 displayed an increase. Half of these proteins (Apolipoprotein A-IV, Cathelicidin antimicrobial peptide, Plasminogen, Retinol-binding protein, and Transthyretin) were associated with the development of early/late IUGR. This underscores the importance of careful group selection and alignment by gestational age during pregnancy studies [35].

Thirteen maternal plasma proteins, mainly apolipoproteins, exhibited differential expression in IUGR and were interconnected. These proteins are involved in plasma lipoprotein assembly, remodeling, and clearance; lipid metabolism (particularly cholesterol and phospholipids); hemostasis (including platelet degranulation); and immune system regulation. The proteins with the highest Cohen’s d effect size included Alpha-2-macroglobulin, Apolipoprotein A-IV, Adiponectin, Apolipoprotein D, Apolipoprotein B-100, and Apolipoprotein(a). The most pronounced changes in Apolipoprotein B-100 were associated with early IUGR, while Apolipoprotein A-IV changes were linked with the late type. This suggests that lipid metabolism and transport are significantly disturbed in IUGR, which is consistent with previous research [36,37,38]. Apolipoproteins have emerged as crucial screening biomarkers for predicting cardiovascular risk, often surpassing traditional serum lipid measurements in predictive power [39]. IUGR may be considered a metabolic disorder, as lipids play a critical role in fetal growth and development. Starting at 24 weeks, there is exponential fetal growth and adipose tissue accumulation due to increased glucose conversion into adipose tissue and fatty acid utilization. During this time, the percentage of adipose tissue in the fetus increases from 3.2% to 16%, which is closely linked to changes in maternal apolipoprotein composition and increased transplacental transport of essential polyunsaturated fatty acids [40,41].

Elevated levels of total cholesterol, triglycerides, and cholesterol-rich low-density lipoprotein (LDL-C), as well as decreased levels of cholesterol-rich high-density lipoprotein (HDL-C), have been associated with increased risks of gestational complications and adverse perinatal outcomes [41,42]. Measurement of serum Apolipoprotein B-100 (apoB100) reflects total LDL-C because each lipoprotein particle contains exactly one molecule of apoB100. Thus, apoB100 is considered a powerful tool for assessing atherogenic lipid status [39]. ApoB100-containing particles transport substantial amounts of cholesterol, triglycerides, and lipophilic vitamins to peripheral tissues, including the placenta and, subsequently, the fetus [41]. Low maternal apoB100 levels, as observed in your study, may reflect inadequate fetal nutrition. A quantitative study by Manja Wolter of 15 maternal plasma apolipoproteins confirmed a statistically significant decrease in apoB100 and its high diagnostic potential (AUC = 0.858) in IUGR cases, including both early and late types of pathology [43]. Furthermore, a study by Lo Vasco VR et al. in 2011 found that fetuses with IUGR displayed intimal/medial thickening (aIMT) of the abdominal aortic wall, potentially indicating an early manifestation of atherosclerosis associated with inflammatory processes, which may serve as an early marker of future vascular changes in adulthood [44].

To our knowledge, it is the first report of a pronounced increase in adiponectin in maternal plasma during IUGR, especially in early onset. Plasma adiponectin is an insulin-sensitizing, anti-inflammatory, and anti-atherogenic adipocytokine primarily secreted by adipocytes. It negatively correlates with insulin resistance [45,46]. During pregnancy, insulin resistance increases, partly due to elevated levels of circulating adipocytokines: TNF-α, resistin, and leptin [47]. As pregnancy advances, insulin synthesis decreases, resulting in increased glucose supply to the growing fetus. The metabolism of pregnant women enters a catabolic phase, with energy referentially obtained through the β-oxidation of free fatty acids in the liver. Adiponectin promotes this process by reducing liver insulin synthesis and adipose tissue lipogenesis while increasing fatty acid oxidation [46]. Both in vivo and in vitro experiments proposed adiponectin-mediated impairment of amino acid transport, leading to a reduction in fetal growth [48,49]. Additionally, adiponectin negatively correlated with newborn cord blood glucose (r = −0.30, *p* = 0.04), cerebroplacental ratio (CPR) (r = −0.36, *p* = 0.02), and positively correlated with uterine artery pulsatility index (UtA PI) (r = 0.42, *p* = 0.004).

Moreover, this study indicates an imbalance between pro-inflammatory and anti-inflammatory proteins in pregnancies with IUGR, favoring a pro-inflammatory innate immune status. One of the most altered processes is the regulation of chemokine production (*p* < 0.001), associated with the proteins LBP, C5, APOD, and ADIPOQ. This abnormality in chemokines has been linked to maternal-fetal immune intolerance, trophoblast dysfunction, and impaired angiogenesis [50,51]. Three members of the alpha macroglobulin superfamily (C5, PZP, and A2M) were upregulated in the IUGR group, with a pronounced tendency to PZP and A2M increase in the early type of disease. A physiological increase in maternal plasma PZP is necessary for the development of feto-maternal immunotolerance [52]. Moreover, PZP stabilizes misfolded proteins [53]. Altered expression of PZP was detected in early pregnancy disorders [54,55,56].

Uterine spiral artery remodeling is essential for proper placental angiogenesis, ensuring adequate nutrition and normal fetal growth. Disruption of these processes increases the risk of developing PE. A2M protein, an inhibitor of antiangiogenic factors (for instance, VEGF and PlGF), was noted to be increased in early PE [57]. In line with our study, a high maternal A2M level was associated with a statistically significant decrease in birth weight in IUGR [58]. Another potential marker of IUGR, Fibronectin, is a hallmark of endothelial damage. The ischemic placenta releases a number of proinflammatory molecules that stimulate systemic inflammation and endothelial damage. Oxidative stress, inflammation, and placental thrombosis may disrupt the placental transfer of nutrients and oxygen to the fetus, restricting its growth [59,60].

In our study, the main differential pathways in early IUGR were associated with the activation of the complement and coagulation cascades and plasma lipoprotein assembly, remodeling, and clearance [61,62]. In accordance with our findings, excessive complement activation is a sign of early severe PE/IUGR and isolated IUGR [63,64]. Additionally, there was a significant increase in the proinflammatory S100 calcium-binding protein A9 (S100A9) in early-onset IUGR. S100A8/A9 modulates the inflammatory response by promoting leukocyte recruitment and inducing cytokine secretion [65]. Incorrect placental development can lead to local ischemia/hypoxia, activation of neutrophils, and the release of proinflammatory factors, including S100A8/A9 [66].

Complement activation is closely linked to increased platelet activity [67,68]. Hypercoagulability is a physiological adaptation of pregnancy to prevent postpartum bleeding. However, with early PE and IUGR, there is a pathological hypercoagulation, which triggers endothelial inflammation and forms a negative feedback loop [69,70,71]. Effective methods for preventing early PE/IUGR include targeting these pathways with low molecular weight heparin or aspirin [72]. Uteroplacental hemostasis during pregnancy depends on fibrinolysis as well. Placenta-derived factors tend to inhibit fibrinolysis [73]. In early IUGR, we observed a significant downregulation of maternal Tissue-type plasminogen activator (PLAT, tPA), involved in the breakdown of blood clots. Notably, the level of PLAT inhibitor, PAI-1 (plasminogen activator inhibitor-1), was not changed (Appendix A). PE, on the other hand, typically shows an increase in both PLAT and PAI-1, indicating systemic endothelial activation [74,75,76,77]. Our study has provided valuable insights into the different patterns of the fibrinolysis system between isolated early-onset IUGR and PE. In the case of PE, pathological platelet activation results in a high level of coagulation, which is partially offset by an overproduction of PLAT and suppression of its inhibitors, PAI-1 and PAI-2. In contrast, fibrinolysis appears to be restricted in early IUGR due to the enhanced plasma level of A2M and downregulation of PLAT. This imbalance in the fibrinolysis system may lead to increased thrombus formation in early IUGR. Unlysed clots can damage the vascular wall, leading to ischemia and infarction of the placenta. Further research in this area may provide valuable insights into the development of targeted interventions and treatments for IUGR to improve pregnancy outcomes.

The majority of IUGR cases (up to 70%) belong to the late-onset. This study reveals that a simultaneous increase in ApoA4 and A2M in maternal plasma can accurately distinguish a subgroup of late-onset IUGR. It is the first documentation of a significant elevation of ApoA4 in IUGR. Chylomicrons and HDLs contain about 25% of plasma ApoA4; the remainder of ApoA4 circulates in free form. ApoA4 regulates lipoprotein metabolism and reverse cholesterol transport, reduces hepatic gluconeogenesis, and possesses anti-inflammatory and antioxidant properties [78,79]. Regarding glucose metabolism, ApoA4 exhibits an effect similar to that of adiponectin, significantly enhanced in early IUGR. Thus, lipid and glucose metabolism are affected in both the late and early stages of IUGR.

ApoA4 also serves as a ligand for platelet integrin αIIbβ3 and has a negative regulatory impact on thrombosis [79]. The synergistic effects of aspirin and apoA4 in treating thrombosis may open new possibilities in therapy [80]. Elevated apoA4 levels observed in late-onset IUGR may protect the mother and fetus, especially during prothrombotic conditions associated with pregnancy. Late-onset IUGR displays the opposite trends to early type with decreased platelet activity and increased fibrinolysis. These findings illuminate the complex interactions between endothelial function and hemostasis in IUGR.

In this study, high diagnostic accuracy (more than 80%) logistic regression models were developed for early and late IUGR diagnosis and for differentiating late IUGR from SGA. Logistic regression, widely used for clinical problems due to its clarity and interpretability, was employed to select variables with maximum diagnostic potential. Selection of variables according to Akaike information criterion (AIC) in combination with other filters (comparative test *p* values, variable importance for projection (VIP) in the orthogonal projections to latent structures discriminant analysis (OPLS-DA), least absolute shrinkage and selection operator (LASSO)) allows to select a set of variables with maximum diagnostic potential [81,82]. The *p*-value is used as an additional parameter to include the independent variable in the model [83,84]. The variables of the logistic regression models included four proteins: Alpha-2-macroglobulin, Apolipoprotein A-IV, Antithrombin-III, and Apolipoprotein C-I, reflecting the main disturbed processes in IUGR, in particular, lipid metabolism and transport, immune response, and blood hemostasis. While various models were previously proposed for diagnosing pregnancy complications, they often lack high diagnostic or prognostic value [9,10]. Therefore, further research into creating models that incorporate new non-invasive/minimally invasive IUGR markers is essential. The prognostic models developed can also be valuable for objective assessments of fetal intrauterine status (e.g., asphyxia). Assessing the risk of neonatal complications (e.g., neonatal hemorrhage) may identify a group of patients who require more careful monitoring in the NICU setting. The use of quantitative analysis methods using HPLC-MS/MS in routine clinical practice has recently begun to be actively implemented in large medical centers and diagnostic laboratories. These approaches are the standard for laboratory diagnostics in terms of accuracy and specificity, and the simultaneous determination of a set of compounds in one analysis makes them cost-effective and competitive.

The current study has a number of strengths, including homogenous gestational age-matched groups, a focus on “pure” IUGR without comorbidities like PE, subdivision into various types of pathology (early, late IUGR, and SGA), antenatal measurements within a narrow gestational time window, and a comprehensive quantitative examination of multiple plasma proteins simultaneously by LC-MRM MS with internal standards. It brings us closer to implementing the developed diagnostic and prognostic models into routine clinical practice. This study has several limitations that should be considered. Firstly, the sample size in this study is relatively small. Secondly, it is a single-center study, which may limit the generalizability of the findings. Early IUGR without concomitant PE is a rare and unique case, making it challenging to draw broad conclusions. To establish the validity and specificity of biomarkers for IUGR and the risk of adverse events in newborns, large-scale, longitudinal, multi-center trials are necessary. These trials will provide more robust evidence and a broader perspective on the utility of the identified biomarkers in clinical practice.

## 4. Materials and Methods

### 4.1. Study Design

A pilot prospective cohort study was conducted, comprising 50 pregnant women who were under observation and received care at the National Medical Research Center for Obstetrics, Gynecology, and Perinatology, named after Academician V.I. Kulakov, of the Ministry of Health of Russia, between 2016 and 2021. Women included in the study provided informed consent.

The inclusion criteria encompassed pregnant women between the ages of 18 and 45 with singleton pregnancies devoid of severe somatic or gynecological pathologies, fetal chromosomal abnormalities or congenital malformations. Antenatal diagnosis of IUGR was based on the Delphi criteria and clinical recommendations of the Ministry of Health of the Russian Federation. Exclusion criteria comprised pre-eclampsia, gestational diabetes mellitus, or any infectious diseases during pregnancy. Women with pregnancies resulting from assisted reproductive technologies were also excluded from the study.

The control group consisted of pregnant women who met the inclusion criteria but did not exhibit clinical or ultrasound signs of IUGR. These women underwent delivery prior to 37 weeks of pregnancy due to premature rupture of amniotic fluid, onset of labor, or scar failure. To ensure the accuracy of the proteome study, late (≥32 weeks) and early (<32 weeks) control groups were based on gestational age at blood sampling.

The study participants were categorized into five groups: group 1—early IUGR (<32 weeks) (n = 10), group 2—late IUGR (≥32 weeks) (n = 10), groups 3 and 4—matched control (n = 10/n = 10), and group 5—(SGA) (≥32 weeks) (n = 10). Blood plasma samples were obtained on the day of delivery.

To confirm the antenatal diagnosis of IUGR and SGA, as well as to validate normal weight in the control groups, postnatally, the weight and length of newborns (n = 50) were assessed using INTERGROWTH-21 centile curves [85,86]. According to international consensus, a child is diagnosed with intrauterine growth restriction if birth weight and/or length is less than the 3rd percentile or if at least 3 of 5 indicators are observed: birth weight is below the 10th percentile; head circumference below the 10th percentile; body length below 10th percentile; prenatally diagnosed fetal growth restriction; complicated pregnancy (arterial hypertension, pre-eclampsia).

### 4.2. Maternal Plasma Collection

Maternal venous blood samples were collected in EDTA tubes prior to delivery and centrifuged at 300× *g* and 4 °C for 20 min. The supernatant was subjected to another round of centrifugation at 12,000× *g* for 10 min. Following these steps, the plasma samples were promptly frozen and stored at −80 °C for further analysis.

### 4.3. Plasma Sample Preparation for Quantitative Analysis

Quantitative analysis of 125 plasma proteins was conducted using the BAK 125 kit (MRM Proteomics Inc, Montreal, Canada) by liquid chromatography-multiple reaction monitoring mass spectrometry (LC-MRM MS). This kit included a stable-isotope-labeled internal standard (SIS) and natural (NAT) synthetic proteotypic peptides to measure the concentration of respective proteins in the blood. All MRM assays in the BAK 125 kit were characterized following the guidelines of the Clinical Proteomic Tumor Analysis Consortium (CPTAC) [87].

The entire process, including sample preparation and LC-MRM MS analysis, followed the manufacturer’s protocol [26,29]. Briefly, plasma samples (10 μL) were denaturated and reduced by incubation with 9 M urea, 20 mM dithiothreitol, and 200 mM Tris×HCl (pH 8.0, +37 °C, 30 min); alkylated by a 30 min incubation in the dark with 100 mM iodoacetamide; diluted with 100 mM Tris×HCl (pH8.0); digested with L-(tosylamido-2-phenyl) ethyl chloromethyl ketone (TPCK)-treated trypsin (Worthington) at a 20:1 (protein:enzyme, *w*/*w*) ratio for 18 h at 37 °C; acidified with 1.0% formic acid (FA); spiked with SIS peptides; cleaned up by solid-phase extraction and reconstituted in 34 μL of 0.1% FA prior to LC-MRM MS analysis.

### 4.4. Quantitative Analysis of 125 Plasma Proteins by LC-MRM MS

Each protein was quantified using a single tryptic peptide to maximize the number of proteins quantifiable in one run. The LC-MRM MS analysis was carried out using an ExionLC™ (UHPLC system (ThermoFisher Scientific, Waltham, MA, USA) coupled to a QTRAP SCI-EX6500+ mass spectrometer (SCIEX, Toronto, ON, Canada). LC-MS parameters were optimized in the previous studies [26,29]. The sample with a volume of 10 µL was HPLC separated by an Acquity UPLC Peptide BEH column (C18, 300 Å, 1.7 µm, 2.1 mm × 150 mm) (Waters, Milford, MA, USA). Gradient elution was performed during 53 min from 2 to 45% of mobile phase B (0.1% formic acid in acetonitrile) with a flow rate 400 µL/min. Mass-spectrometry analysis was performed with electrospray ionization (4 kV, 450 °C, 40 L/min) and MRM acquisition method (Appendix A) [87].

The Skyline Quantitative Analysis software (version 20.2.0.343, University of Washington) was employed for the visual examination of the LC-MRM MS data. Manual assessment of chromatographic peaks for NAT and SIS peptides in the samples, calibration curves, and quality control (QC) included peak selection, peak shape, and integration accuracy. Calibration curves were obtained using 1/x^2^ weighted linear regression. Calculated peptide concentrations in samples were expressed as fmol/μL plasma. Values below the lower limit of quantification (LLOQ) were substituted with a random number within the range (0; LLOQ). Appendix A provides details on MRM transitions (Appendix A) and an exemplary extracted ion chromatogram of MRM transitions (Appendix A).

### 4.5. Statistical Data Processing

Statistical analysis was conducted using the following software programs and scripts: Statistica 12.6, IBM SPSS Statistics 21, and custom scripts developed in the R 4.2.1 language. Additional plugin libraries ropls, e1071, and pROC were utilized.

The normality of quantitative indicators was assessed through the Shapiro–Wilk test (for sample sizes less than 50). Descriptive statistics were applied for normally distributed data, including means (M), standard deviations (SD), and 95% confidence intervals (95% CI). Non-normally distributed data were described using median (Me) values and interquartile ranges (IQR) [Q1; Q3]. Categorical data were presented using absolute values and percentages.

Comparisons among three or more groups for normally distributed quantitative variables were performed using one-way analysis of variance (ANOVA), followed by post-hoc comparisons utilizing the Tukey test (in case of equal variances) or the Games-Howell test (unequal variances). For non-normally distributed quantitative variables, the Mann–Whitney U test was applied for comparisons between two groups, while the Kruskal–Wallis test was used for comparisons among three or more groups, followed by post-hoc comparisons with Dunn’s test incorporating Holm’s correction. A comparison of percentages in multifield contingency tables was executed using the Pearson chi-square test. Statistical significance was established at a level of *p* < 0.05. For diagnostic problem-solving involving small sample sizes (n = 10 for each group) in “early IUGR vs. early control,” “late IUGR vs. late control,” and “late IUGR vs. SGA” scenarios, models were developed based on logistic regression. These models considered protein concentrations to the first and second powers and the product of protein concentrations. In addressing the prognostic issue of “newborn bleeding,” logistic regression models were created considering protein concentrations to the first power and their ratios. These models were constructed following preliminary data processing [88,89].

The logistic regression models incorporated interactions between variables and had the form y = 1/(1 + e^−x × trans(β)^), with x representing a vector of independent variables in the form {C_i_, C_i_ ×C_j_, C_i_^2^} or {C_i_, C_i_/C_j_}, where C_i_ signified the concentration of the i-th protein, and β denoted the vector of coefficients. A new dataset was formed that included protein concentrations (C_i_), products of the concentrations of each protein pair (C_i_ × C_j_), the squared values of each protein’s concentration (C_i_^2^) or protein concentrations (C_i_) and ratios of the concentrations of each protein pair (C_i_/C_j_). From this dataset, variables with variable importance for projection (VIP) in the orthogonal projections to latent structures discriminant analysis (OPLS-DA) exceeding one were selected, followed by a stepwise exclusion to minimize the Akaike information criterion until no further reduction was observed. Subsequently, variables were eliminated step-by-step, with coefficients reaching a probability of being equal to zero, less than 0.05. Model quality assessment was conducted through cross-validation for individual objects and sensitivity and specificity calculations, maximizing the sum of sensitivity and specificity based on the model’s results.

Protein–protein interactions were analyzed using the STRING database (accessed on 6 January 2023). Only associations with *p* < 0.05 were included in the final networks. Protein categorical annotations were derived from GeneOntology via the SwissProt database.

## 5. Conclusions

The prenatal identification of infants at a heightened risk of small for gestational age (SGA), intrauterine growth restriction (IUGR), or adverse perinatal outcomes holds paramount importance. It has the potential to inform decisions regarding the timing and mode of delivery, enabling the provision of effective antenatal care. In this study, we present the first results for multiplexed absolute quantitation of 125 plasma of maternal circulating proteins and their associated pathophysiological pathways, offering potential molecular pathways disturbed in isolated IUGR.

Thirteen potential markers of IUGR (Gelsolin, Alpha-2-macroglobulin, Apolipoprotein A-IV, Apolipoprotein B-100, Apolipoprotein(a), Adiponectin, Complement C5, Apolipoprotein D, Alpha-1B-glycoprotein, Serum albumin, Fibronectin, Glutathione peroxidase 3, Lipopolysaccharide-binding protein) were found to be interconnected in a protein–protein network. These proteins participate in plasma lipoprotein assembly, remodeling, and clearance; lipid metabolism, especially cholesterol and phospholipids; hemostasis, including platelet degranulation; and immune system regulation. Additionally, 18 proteins were specific to a particular type of IUGR (early or late). Early IUGR was associated with disturbances in the activation of the complement/coagulation cascades and plasma lipoprotein assembly, remodeling, and clearance, whereas late IUGR primarily affected vitamin digestion/absorption and responses to lipid hydroperoxide pathways. Moreover, distinct patterns in the coagulation and fibrinolysis systems were observed between isolated early- and late-onset IUGR. Specifically, late-onset IUGR exhibited trends opposite to those of the early type, characterized by reduced platelet activity and increased fibrinolysis.

Furthermore, diagnostic models for IUGR and SGA were proposed, incorporating Alpha-2-macroglobulin, Apolipoprotein A-IV, Antithrombin-III, and Apolipoprotein C-I as variables. Logistic regression models were also developed to predict fetal/newborn asphyxia and neonatal hemorrhage. Accurate and timely diagnosis of IUGR and assessment of the risk of asphyxia can optimize management strategies, including decisions related to the timing and mode of delivery. More vigilant monitoring in the NICU setting is necessary for infants with an increased risk of neonatal complications.

In conclusion, our findings highlight the complex interplay of immune and coagulation factors in IUGR and the differences between early- and late-onset IUGR and other placenta-related conditions like PE. Understanding these mechanisms is crucial for developing targeted interventions and improving outcomes for pregnancies affected by IUGR.

## Figures and Tables

**Figure 1 ijms-24-16832-f001:**
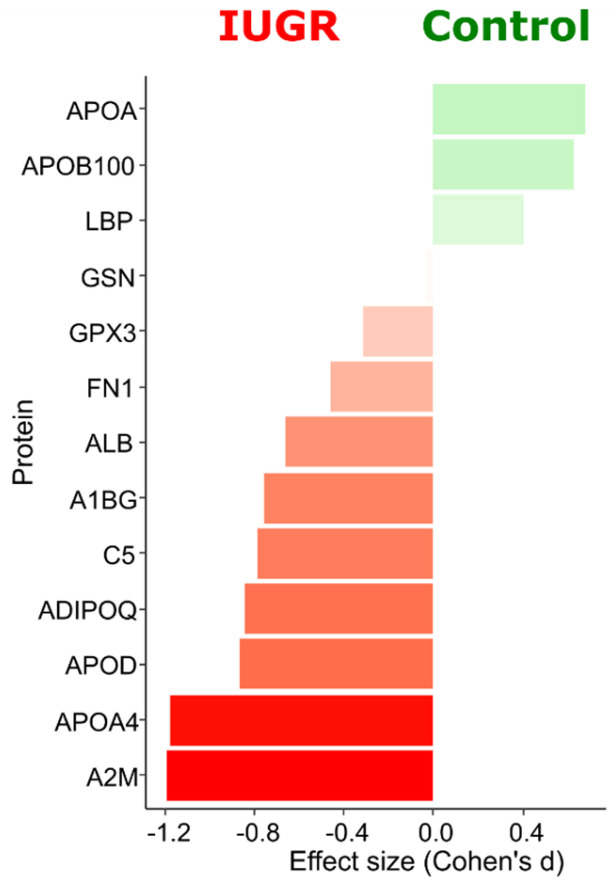
The effect sizes (Cohen’s d) ordering of 13 proteins statistically different between the control and IUGR groups. GSN—Gelsolin, A2M—Alpha-2-macroglobulin, APOA4—Apolipoprotein A-IV, APOB100—Apolipoprotein B-100, APOA—Apolipoprotein(a), ADIPOQ—Adiponectin, C5—Complement C5, APOD—Apolipoprotein D, A1BG—Alpha-1B-glycoprotein, ALB—Serum albumin, FN1—Fibronectin, GPX3—Glutathione peroxidase 3, LBP—Lipopolysaccharide-binding protein.

**Figure 2 ijms-24-16832-f002:**
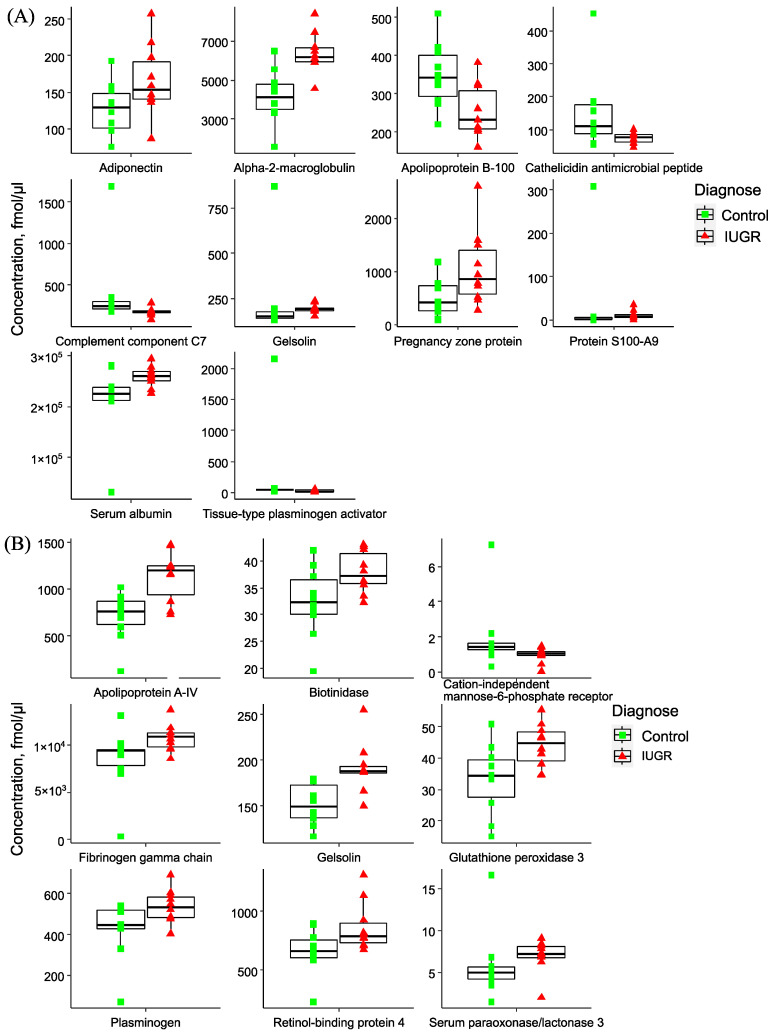
Boxplots of plasma proteins with significantly different protein concentrations in (**A**) early and (**B**) late IUGR, compared to relevant control group. The median is presented as a horizontal line in the interquartile range box with minimum and maximum whiskers. Individual data points are shown as green squares (control) and red triangles (IUGR).

**Figure 3 ijms-24-16832-f003:**
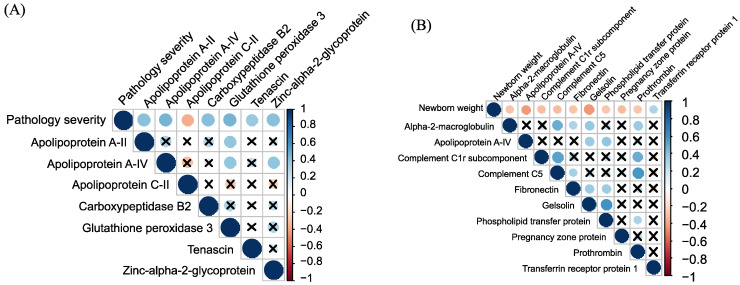
Corrplot of maternal plasma proteins correlated significantly with the pathology severity (1—SGA, 2—late IUGR, 3—early IUGR) (**A**) and newborn weight (**B**).

**Figure 4 ijms-24-16832-f004:**
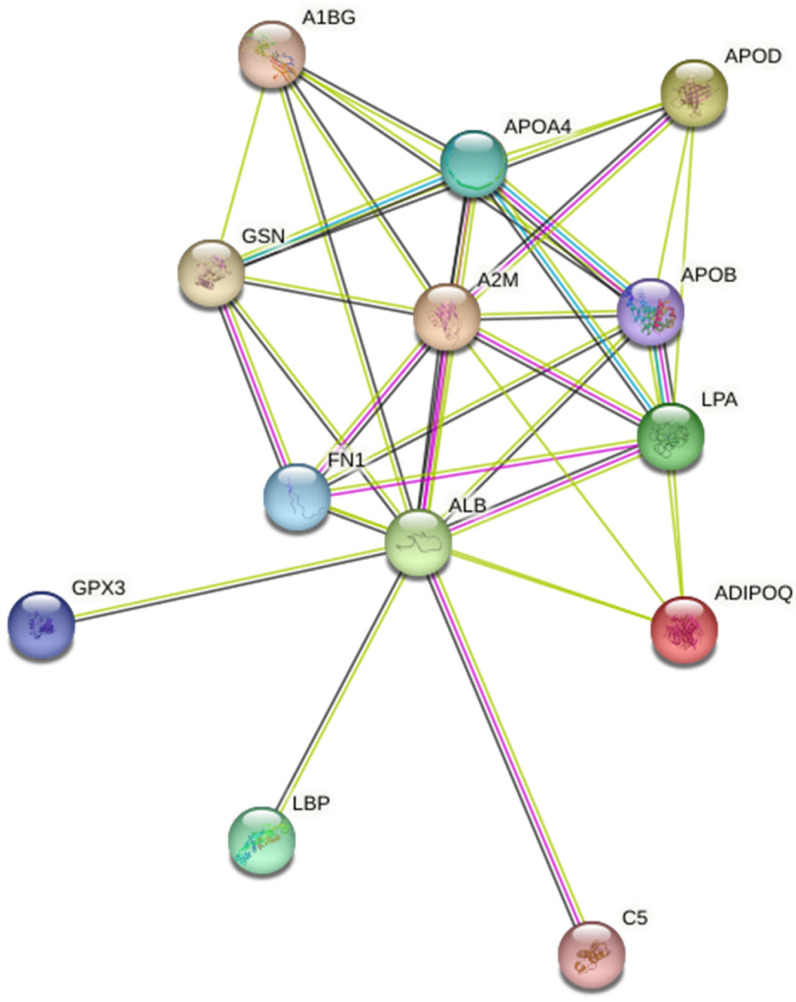
Protein–Protein Interaction network for 13 differentially expressed plasma protein in IUGR obtained using STRING database 11.00 (https://string-db.org/, accessed on 6 January 2023). Nodes represent proteins and edges represent protein–protein associations: purple indicates experimentally determined interactions, blue indicates interactions from the curated database, black indicates co-expression of genes and green indicates text mining—the result of parsing full-text articles from the PMC Open Access Subset, PubMed abstracts, summary texts from OMIM (OMIM.org) and SGD (Saccharomyces Genome Database) entry descriptions. GSN—Gelsolin, A2M—Alpha-2-macroglobulin, APOA4—Apolipoprotein A-IV, APOB100—Apolipoprotein B-100, APOA—Apolipoprotein(a), ADIPOQ—Adiponectin, C5—Complement C5, APOD—Apolipoprotein D, A1BG—Alpha-1B-glycoprotein, ALB—Serum albumin, FN1—Fibronectin, GPX3—Glutathione peroxidase 3, LBP—Lipopolysaccharide-binding protein.

**Figure 5 ijms-24-16832-f005:**
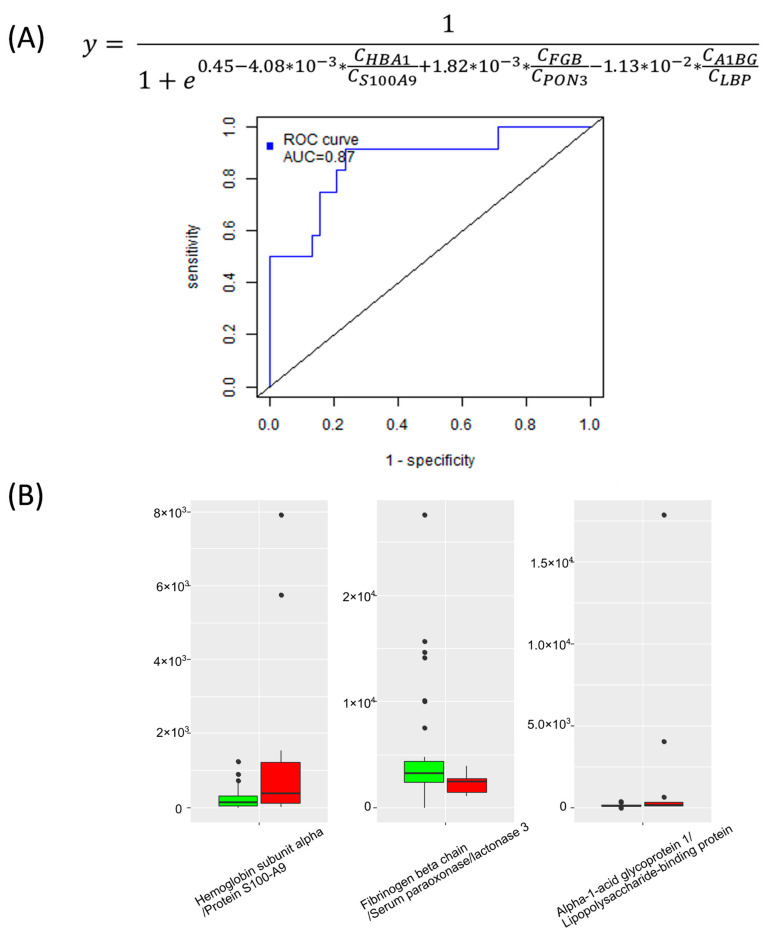
ROC curve with the logistic regression formula (**A**) and boxplots (**B**) of variables used in the model for predicting the newborn bleeding in the early neonatal period. Red color—development of bleeding, green color—absence of bleeding.

**Table 1 ijms-24-16832-t001:** Demographic and clinical data of pregnant women and fetuses.

	Group 1, Early IUGR (*n* = 10)	Group 2, Late IUGR (*n* = 10)	Group 3, Early Control (<32 wks) (*n* = 10)	Group 4, Late Control (≥32 wks) (*n* = 10)	Group 5, SGA (*n* = 10)	*p*-Value
Age, years M ± SD (95% Cl)	34 ± 6 (30–38)	31 ± 5 (27–34)	30 ± 4 (27–33)	30 ± 3 (28–33)	32 ± 4 (28–35)	≥0.05
BMI M ± SD (95% CI)	24 ± 3 (22–26)	24 ± 3 (22–26)	28 ± 6 (24–32)	27 ± 5 (24–31)	26 ± 5 (22–30)	≥0.05
IUGR onset, weeks M ± SD (95% CI)	25 ± 3 (23–27)	34 ± 2 (33–35)	-	-	35 ± 2 (34–36)	<0.05 * *p*_1/2_ < 0.001 *p*_1/5_ < 0.001
Ultrasound-based fetal weight estimation, percentile Me [Q_1_;Q_3_;]	2 [1;3]	1 [0;2]	43 [21;66]	59 [45;74]	2 [1;4]	<0.05 * *p*_1/3_ = 0.012 *p*_1/4_ = 0.001 *p*_2/3_ = 0.001 *p*_2/4_ < 0.001 *p*_3/5_ = 0.025 *p*_4/5_ = 0.004
Uterine artery pulsatility index (UtA PI), left, percentile Me [Q_1_;Q_3_;]	88 [76;99]	97 [78;99]	47 [42;78]	88 [66;94]	65 [53;74]	≥0.05
Uterine artery pulsatility index (UtA PI), right, percentile Me [Q_1_;Q_3_;]	91 [62;100]	95 [55;99]	71 [42;89]	94 [75;98]	58 [36;79]	≥0.05
Umbilical artery pulsatility index (UA PI), percentile Me [Q_1_;Q_3_;]	99 [84;100]	83 [54;97]	67 [52;75]	68 [27;87]	52 [37;68]	≥0.05
Cerebroplacental ratio (CPR), percentile Me [Q_1_;Q_3_;]	1 [1;2]	16 [2;39]	33 [27;42]	62 [36;67]	54 [34;60]	<0.05 * *p*_1/5_ = 0.026 *p*_1/4_ = 0.004
Fetal middle cerebral artery pulsatility index (MCA PI), percentile Me [Q_1_;Q_3_;]	12 [2;18]	17 [3;47]	28 [19;41]	89 [52;96]	49 [31;59]	<0.05 * *p*_1/4_ = 0.006
Delivery time, days Me [Q_1_;Q_3_;]	217 [208;250]	260 [253;268]	260 [227;262]	276 [250;282]	267 [264;270]	<0.05 * *p*_1/5_ = 0.008 *p*_1/4_ = 0.003

* statistically significant differences between groups (*p* < 0.05).

**Table 2 ijms-24-16832-t002:** Diagnostic performance (AUC, sensitivity and specificity) and variables of the logistic regression models for early/late IUGR and SGA. Coefficient β, confidence interval CI β, Wald criteria Z, and coefficient zero-probability P are provided.

	Sensitivity	Specificity	AUC	Variables	β	CI β	Z	P
**Early IUGR vs. early control**	0.9	0.9	0.86	Intercept	−5.08	−11.04–−1.66	−2.27	0.02
				Alpha-2-macroglobulin^2^	1.71 × 10^−7^	6.15 × 10^−8^–3.54 × 10^−7^	2.42	0.02
**Late IUGR vs. late control**	0.9	0.8	0.88	Intercept	−7.68	−17.89–−2.37	−2.02	0.04
				Alpha-2-macroglobulin * Apolipoprotein A-IV	1.45 × 10^−6^	4.55 × 10^−7^–3.39 × 10^−6^	1.98	0.047
**Late IUGR vs. SGA**	0.8	0.8	0.8	Intercept	5.31	1.36–11.39	2.15	0.03
				Antithrombin-III * Apolipoprotein C-I	−2.89 × 10^−7^	−6.20 × 10^−7^–−7.80 × 10^−8^	−2.15	0.03

## Data Availability

Data are contained within the Appendix A.

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
