# Peer review of "Quantitative Proteomics of Maternal Blood Plasma in Isolated Intrauterine Growth Restriction"

_ijms, 2023, doi:10.3390/ijms242316832_

Round 1
Reviewer 1 Report
Comments and Suggestions for Authors
In the current study the authors aimed to investigate the disturbance in biological processes associated with isolated IUGR via blood plasma proteomics. They described a complex interplay of immune and coagulation factors in IUGR and the differences between early and late-onset IUGR.
The study design is clear and the manuscript is well written. The methods and results are described in details, the figures are clear and appropriated.
My minor comment: The authors are encouraged to discuss how this approach maight be introduced in the clinical practice for early dignosis also in light of a cost-effectiveness evaluation
Comments on the Quality of English Language
Minor editing of English language required
Author Response
In the current study the authors aimed to investigate the disturbance in biological processes associated with isolated IUGR via blood plasma proteomics. They described a complex interplay of immune and coagulation factors in IUGR and the differences between early and late-onset IUGR.
The study design is clear and the manuscript is well written. The methods and results are described in details, the figures are clear and appropriated.
Minor comment: The authors are encouraged to discuss how this approach maight be introduced in the clinical practice for early dignosis also in light of a cost-effectiveness evaluation
Answer. Thank you for reviewing our article. The following text was added in discussion: “The use of quantitative analysis methods using HPLC-MS/MS in routine clinical practice has recently begun to be actively implemented in large medical centers and diagnostic laboratories[Gelb, M.H.; Basheeruddin, K.; Burlina, A.; Chen, H.-J.; Chien, Y.-H.; Dizikes, G.; Dorley, C.; Giugliani, R.; Hietala, A.; Hong, X.; et al. Liquid Chromatography–Tandem Mass Spectrometry in Newborn Screening Laboratories. Int. J. Neonatal Screen. 2022, 8, 62. https://doi.org/10.3390/ijns8040062; Gaither C, Popp R, Mohammed Y, Borchers CH. Determination of the concentration range for 267 proteins from 21 lots of commercial human plasma using highly multiplexed multiple reaction monitoring mass spectrometry. Analyst. 2020;145(10):3634‐3644]. These approaches are the standard for laboratory diagnostics in terms of accuracy and specificity, and the simultaneous determination of a set of compounds in one analysis makes them cost-effective and competitive.”.
Reviewer 2 Report
Comments and Suggestions for Authors
The main question addressed by the research is deriving from the title of the article “Quantitative Proteomics of Maternal Blood Plasma in Isolated Intrauterine Growth Restriction”. In other words, the main question addressed by the research is if there is any disturbance in plasma proteomics associated with isolated FGR.
Compared with other published material, the article examines the disturbance in plasma proteomics associated with isolated FGR (differing from other published material examining this relation in other pathologic conditions or in the co-existence of FGR with other pathology of pregnancy). The term “isolated” (idiopathic) in the title is self-explaining what it adds to the subject area compared with other published material.
Even though the paper borrows and even improves on the wording of the source (King et al. 2022), it is not clear how growth potential is determined, nor is a clear threshold defined; according to this definition, we would still be dealing with IUGR even if the fetus reached 99% of its growth potential. Considering that the contemporary term is "fetal growth restriction" (FGR) which is determined as an estimated fetal weight or abdominal circumference <10th percentile for gestational age (GA), with severe FGR defined as an estimated fetal weight or abdominal circumference <3rd percentile for GA, please consider adjusting the rest of the text accordingly.
Author Response
The main question addressed by the research is deriving from the title of the article “Quantitative Proteomics of Maternal Blood Plasma in Isolated Intrauterine Growth Restriction”. In other words, the main question addressed by the research is if there is any disturbance in plasma proteomics associated with isolated FGR.
Compared with other published material, the article examines the disturbance in plasma proteomics associated with isolated FGR (differing from other published material examining this relation in other pathologic conditions or in the co-existence of FGR with other pathology of pregnancy). The term “isolated” (idiopathic) in the title is self-explaining what it adds to the subject area compared with other published material.
Even though the paper borrows and even improves on the wording of the source (King et al. 2022), it is not clear how growth potential is determined, nor is a clear threshold defined; according to this definition, we would still be dealing with IUGR even if the fetus reached 99% of its growth potential. Considering that the contemporary term is "fetal growth restriction" (FGR) which is determined as an estimated fetal weight or abdominal circumference <10th percentile for gestational age (GA), with severe FGR defined as an estimated fetal weight or abdominal circumference <3rd percentile for GA, please consider adjusting the rest of the text accordingly.
Answer. Thank you for your kind opinion on our research. A study by King et al. 2022 [1] was cited as an example of one of the up-to-date antenatal definitions of fetal growth restriction. In our study, confirmation of the diagnosis of FGR and division into groups was carried out after postnatal assessment of anthropometric parameters in newborn children in the studied women according to the INTERGROWTH-21st centile curves [2,3]. According to international consensus, a child is diagnosed with intrauterine growth restriction if birth weight and/or length is less than the 3rd percentile or if at least 3 of 5 indicators are observed: birth weight is below the 10th percentile; head circumference below the 10th percentile; body length below 10th percentile; prenatally diagnosed fetal growth restriction; complicated pregnancy (arterial hypertension, preeclampsia). The corresponding text has been added to the Materials and Methods section.
- King VJ, Bennet L, Stone PR, Clark A, Gunn AJ and Dhillon SK (2022), Fetal growth restriction and stillbirth: Biomarkers for identifying at risk fetuses. Physiol. 13:959750. doi: 10.3389/fphys.2022.959750
- Leite D.F.B., de Melo E.F. Jr, Souza R.T., Kenny L.C., Cecatti J.G. Fetal and neonatal growth restriction: new criteria, renew challenges // J. Pediatr. 2018. Vol. 203. P. 462–463. DOI:https://doi.org/10.1016/j.jpeds.2018.07.094 Epub 2018 Aug 29. PMID: 30172439.
- Beune IM, Bloomfield FH, Ganzevoort W, Embleton ND, Rozance PJ, van Wassenaer-Leemhuis AG, Wynia K, Gordijn SJ. Consensus Based Definition of Growth Restriction in the Newborn. J Pediatr. 2018 May;196:71-76.e1. doi: 10.1016/j.jpeds.2017.12.059. Epub 2018 Feb 28. PMID: 29499988.